# Analysis of the Difference between Climate Aridity Index and Meteorological Drought Index in the Summer Monsoon Transition Zone

**Hongli Zhang [1], Liang Zhang [2], Qiang Zhang [2,\*], Qian Liu [3], Xiaoni You [1] and Lixia Wang [1]**

1   College of Resources and Environmental Engineering, Tianshui Normal University, Tianshui 741001, China
2   Institute of Arid Meteorology, CMA, Key Laboratory of Arid Climatic Change and Reducing Disaster of Gansu Province, Key Laboratory of Arid Climatic Change and Disaster Reduction of CMA, Lanzhou 730020, China
3   School of Atmospheric Sciences, Sun Yat-sen University, Zhuhai 519082, China
\*   Correspondence: zhangqiang@cma.gov.cn

**Abstract:** The summer monsoon transition zone (SMTZ) in China represents an unusual land type with an agro-pasture ecotone, and it is a climate-sensitive region. Changes in climate aridity and changes in meteorological drought are mutually related yet fundamentally different. In this study, potential evapotranspiration ($E_{TO}$) is calculated using Penman–Monteith, based on China's national meteorological stations data from 1961 to 2013. An $E_{TO}$-based climate aridity index ($I_{AI}$) and $E_{TO}$-based standardized precipitation evapotranspiration index (SPEI) are used as the metrics for climate aridity and meteorological drought, respectively. The result shows a significant difference between climate aridity and meteorological drought in the SMTZ, compared with the monsoon and non-monsoon zone. This difference varies on different time scales (1–48 months), and the greatest differences between $I_{AI}$ and SPEI are on seasonal and monthly scales (1–12 months), but lower at longer time scales (>12 months). The first reason for the difference is the desynchronicity of meteorological drought and the background climate. After the background climate becomes a relatively arid state (such as $0.96 < I_{AI} < 1$) from a semi-arid state ($0.50 < I_{AI} < 0.80$), the continued arid state with weak $I_{AI}$ fluctuations eventually results in increasingly severe meteorological droughts, or the recurrence of equally severe droughts with drastic reduction. Consequently, the onset of the most severe climate aridity is two to seven months (mostly three to four months) ahead of the onset of the most severe drought events, until the climate returns to a semi-arid state. Second, climate aridity represents the average state of the background climate over a long time period and changes gently, while meteorological droughts are stochastic climate events and change drastically. These findings indicate that $I_{AI}$ can serve as a predictor of the onset of meteorological drought events, especially in the SMTZ, but it fails to characterize the progression of meteorological drought events well. Therefore, this result is of great significance for drought prediction and early warning.

**Keywords:** summer monsoon transition zone; dryness and wetness of background climate; meteorological drought; aridity

## 1. Introduction

The Chinese summer monsoon transition zone (SMTZ) has a primarily semi-arid climate [1] and it is a geographical transition region between humid and arid climates zones. Transition regions such as the SMTZ are sensitive to changes in climate and are subject to frequent droughts. In these regions, the accurate prediction of precipitation changes in the background climate and meteorological drought monitoring can improve drought prevention capabilities.

Climate aridity reflects the dryness of a regional climate. Typically, climate aridity is measured using the ratio of potential evapotranspiration to actual precipitation, also known

as the aridity index ($I_{AI}$); the reciprocal of this ratio is often defined as the surface moisture index, with AI and the relative moisture index (M; M = surface moisture index−1) usually adopted as metrics to describe the climatic background state [2,3]. In contrast, meteorological drought refers to a water shortage that results from an imbalance between precipitation and evaporation. A meteorological drought may be of short or long duration, and can also refer to a multi-year period during which precipitation is considerably less than the climate mean. Changes in meteorological drought usually take place on shorter time scales than changes in climate aridity, and are typically evaluated using meteorological drought indices such as the precipitation anomaly percentage [4], standardized precipitation index (SPI) [5], standardized precipitation evapotranspiration index (SPEI) [6,7], Palmer Drought Index (PDSI) [8], and Reconnaissance Drought Index (RDI) [9]. Although these indices vary in specifics, each provides a reasonable measure of meteorological drought [10,11]. A composite meteorological drought index (CI), which combines aspects of the indices currently in use, has been proposed and used in China [12].

Occurrences of anomalous climate aridity and meteorological drought are directly determined by precipitation ($P_{RE}$) and potential evapotranspiration ($E_{TO}$), with precipitation as the most important factor. The $E_{TO}$ is affected by thermodynamic and dynamic variables such as temperature, radiation, and wind speed. As similar factors regulate climate aridity and meteorological drought, the evolution of the two quantities is often connected. However, climate aridity and meteorological drought display different variations on different time scales, even in response to the same meteorological factors.

Changes in climate aridity and drought due to anthropogenic climate change have been of great concern in the academic community [13,14]. When investigating inter-annual and longer time scale changes in drought, some studies employ the climate aridity index while others use meteorological drought indices. Studies that use climate aridity indices find that the global arid area is expanding, with the most significant aridification occurring in semi-arid regions [15–17]. Studies using PDSI, a meteorological drought index, have found a similar global aridification trend [18], though this trend may be overestimated due to how the PDSI models potential evapotranspiration [19]. China-based studies on long-term changes in the moisture index, aridity index, PDSI, SPI, and SPEI have shown a drying trend in China, particularly in Northern China [20–23]. Within China, the PDSI index and surface moisture index appear capable of accurately describing the nature and intensity of drought on inter-annual time scales [24,25]. Moreover, the changes in the two indices are consistent with the changes in extreme drought events in northern China [26]. The similar conclusions reached by studies employing climate aridity indices and drought indices indicate that the two types of indices are consistent on inter-annual and longer time scales.

Similarly, some studies still use the index (surface humidity index or relative humidity index) which essentially represents climate dryness as the drought index to study drought on the seasonal and shorter monthly scales. Wang et al. [27] used the relative moisture index to investigate the temporal and spatial distribution of seasonal drought events in southwestern China; Zhou et al. [28] used aridity to define the drought level and investigated meteorological and climate characteristics during drought periods; Ma et al. [26] also used the relative moisture index to measure changes in northeastern China drought trends during May–September from 1961 to 2009. Yao et al. [29] used the relative moisture index as a spring drought indicator to study drought in southwestern China; Huang et al. [17] used the moisture index to calculate the frequency of extreme drought in northwestern China. However, is it reliable to study drought by using the index that essentially represents climate dryness and wetness? However, few studies have been conducted to compare the two types of index, and to analyze their differences on different time scales is an important scientific issue that should be addressed in studies of arid climates and drought events.

China's SMTZ represents an unusual land type with an agro-pasture ecotone, and is a transition zone between humid and arid climates. The SMTZ is subject to frequent drought events, and has been shown to be sensitive to climate changes [15,30,31]. The weather and

climate of China's SMTZ are closely related to the East Asian summer monsoon system. Therefore, analyzing the differences between climate aridity changes and meteorological drought changes in the SMTZ, and comparing the differences with those in monsoon and non-monsoon zones, will facilitate a comprehensive, multi-perspective analysis of changes in regional aridity and drought. These results will provide an improved theoretical basis for drought monitoring in China's SMTZ.

## 2. Materials and Methods

### 2.1. Data

Data used in this study were collected from 785 China Meteorological Administration monitoring stations, and included precipitation, maximum and minimum daily air temperature, relative humidity, wind speed, and daylight hours. To ensure data homogeneity, data selection was based on the following criteria: if a station was missing daily data for at least two days in a month, that month was considered to be a data-lacking month for that station; if a station had five or more data-lacking months during any year, that station was excluded from this study. Using these selection criteria, 661 stations were selected and the data from every station used are available with a monitoring period from 1961 to 2013.

China is a typical summer monsoon region, which can be divided into a summer monsoon zone, an SMTZ, and a non-summer-monsoon zone based on the monsoon impact in each region [32]. As the west of the semi-arid area at 100°E is located in the Tibet Plateau, which is mainly affected by the South Asian high, while the east of the semi-arid area at 100°E is almost located at the northern edge of the East Asian summer monsoon (mainly the southwest and southeast summer monsoon), the east of the semi-arid area ($0.5 < I_{AI} < 0.8$) at 100°E is selected as the representative area of the SMTZ. Given that the climate of the semi-arid zone varies greatly with longitude, we used rotated empirical orthogonal function analysis of $I_{AI}(1-P_{RE}/E_{TO})$ to divide the zone into three sub-zones (A, B, and C) for research [21] (Figure 1). Furthermore, to better understand the relationship between climate aridity and drought occurrences in the SMTZ, two additional zones, a monsoon zone (D) and a non-monsoon zone (E), are included in this study (Figure 1). Given that the Chinese summer monsoon has the most significant precipitation impacts in the middle and lower reaches of the Yangtze River, stations in zone D were primarily selected from extremely humid regions ($-0.4 < I_{AI} < 0.2$). Stations in zone E were primarily in the arid regions of northwestern China ($I_{AI} > 0.95$). The climate aridity and station distribution in each zone are shown in Figure 1 and Table 1.

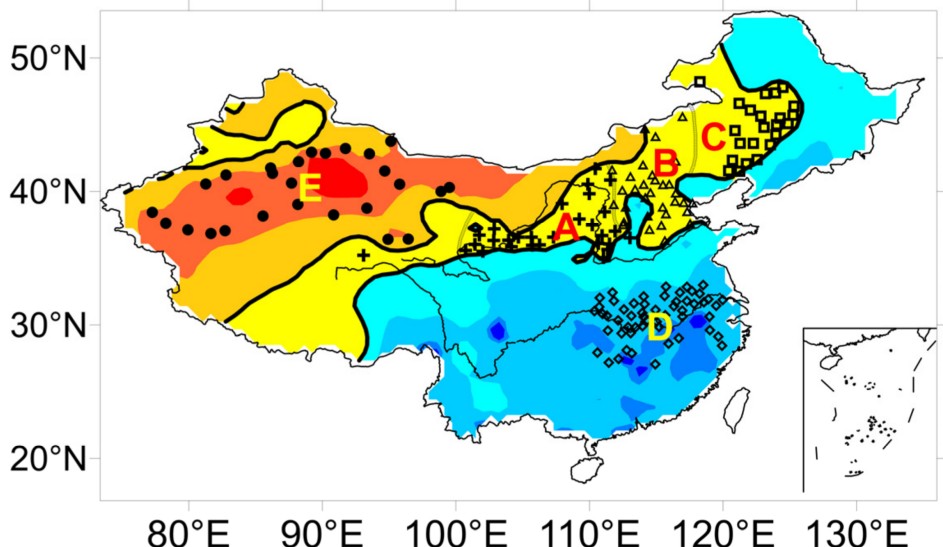

**Figure 1.** Distribution of representative monitoring stations in the summer monsoon transition zone (**A**–**C**), monsoon zone (**D**), and non-monsoon zone (**E**).

**Table 1.** Annual mean climate aridity and number of stations in each of the five regions.

| Region | A (Semi-Arid) | B (Semi-Arid) | C (Semi-Arid) | D (Extremely Humid) | E (Extremely Arid) |
|---|---|---|---|---|---|
| $I_{AI}$ | 0.5–0.8 | 0.5–0.8 | 0.5–0.8 | −0.4–0.2 | >0.95 |
| Number of stations | 37 | 30 | 31 | 55 | 26 |

*2.2. Methods*

Climate aridity and meteorological drought both have varying definitions. Some indices are defined from precipitation alone, while other indices incorporate thermal and dynamical factors. In a warming climate, using precipitation data alone is insufficient to fully describe changes in climate aridity and drought. Recent research has increasingly focused on the effects of warming-enhanced potential evaporation on the changes in climate aridity and drought [33]. The $E_{TO}$ refers to the regional evapotranspiration capacity that would occur given an unlimited surface water supply and would be influenced by air temperature, wind speed, surface net radiation, actual vapor pressure, saturation vapor pressure, and soil heat flux. With climate warming, increasing air temperature has significant impacts on climate aridity and meteorological drought. Therefore, this study selects and compares a climate aridity index and a meteorological drought index that each incorporates $E_{TO}$.

A climate aridity index is an indicator of the degree of dryness in a region, and is measured using the water budget and heat balance. Since 1990, 22 methods to calculate climate aridity have been proposed for calculating, with many of them based on $E_{TO}$ [34]. Traditionally, climate aridity is defined as the ratio of $E_{TO}$ to precipitation [21]. However, when calculating monthly aridity, the denominator, monthly precipitation, may be zero for some months. To avoid this problem, this study defines the aridity index as:

$$I_{AI} = \frac{E_{TO} - P_{RE}}{E_{TO}}, \tag{1}$$

$I_{AI}$ is climate dryness index, $P_{RE}$ is precipitation, and $E_{TO}$ is potential evapotranspiration. A larger value of $I_{AI}$ indicates a drier climate. It is possible to use the accumulated precipitation and $E_{TO}$ to calculate climate aridity on varying time scales. For example, when calculating $I_{AI}$ on a one-month time scale, the $I_{AI}$ value of month *i* is calculated using the precipitation and $E_{TO}$ of month *i*; when calculating $I_{AI}$ on a two-month time scale, the $I_{AI}$ value of month *i* is calculated using the cumulative precipitation and $E_{TO}$ over the period of month $i - 1$ to month *i*. $E_{TO}$ is calculated using the revised Penman–Monteith (PM) model introduced in 1998 by the United Nations Food and Agriculture Organization [35]. The PM model, which is based on a solid theoretical foundation with clear physical meaning, has become a commonly used method for estimating $E_{TO}$. When using the model to calculate $E_{TO}$ in China, relevant parameters are modified according to measured radiation data [36] and the modified model is expressed as follows:

$$, E_{TO} = \frac{0.408\Delta(R_n - G) + \gamma\frac{900}{T+273}U_2(e_a - e_d)}{\Delta + \gamma(1 + 0.34U_2)} \tag{2}$$

where

$$R_n = 0.77 \times (0.2 + 0.79(\frac{n}{N}))Rso - \sigma(\frac{T_{x,k}^4 + T_{n,k}^4}{2}) \times (0.56 - 0.25\sqrt{e_a})(0.1 + 0.9(\frac{n}{N})) \tag{3}$$

where $R_n$ is the surface net radiation (MJ m$^{-2}$ d$^{-1}$); $G$ is the soil heat flux (MJ m$^{-2}$ d$^{-1}$); $e_a$ and $e_d$ are the saturation vapor pressure and actual vapor pressure, respectively (kPa); $\Delta$ is the slope of the plot of vapor pressure versus air temperature (kPa K$^{-1}$); $\gamma$ is the psychrometric constant (kPa K$^{-1}$); $U_2$ is the wind speed at 2 m height (m s$^{-1}$); and *t* is the mean air temperature (°C).

Although $E_{TO}$ is not equal to the evaporation from an evaporation pan (EPAN), they both reflect the evaporation capacity under specific climatic conditions and are well correlated with each other. As EPAN is available at 53 monitoring stations in China, it is possible to use EPAN to test the accuracy of estimated $E_{TO}$ in China, as calculated from Equation (3). As shown in Figure 2, the inter-annual correlation coefficient between EPAN and $E_{TO}$ was statistically significant ($p < 0.01$) at all monitoring stations except one, indicating that the estimated $E_{TO}$ value was highly reliable.

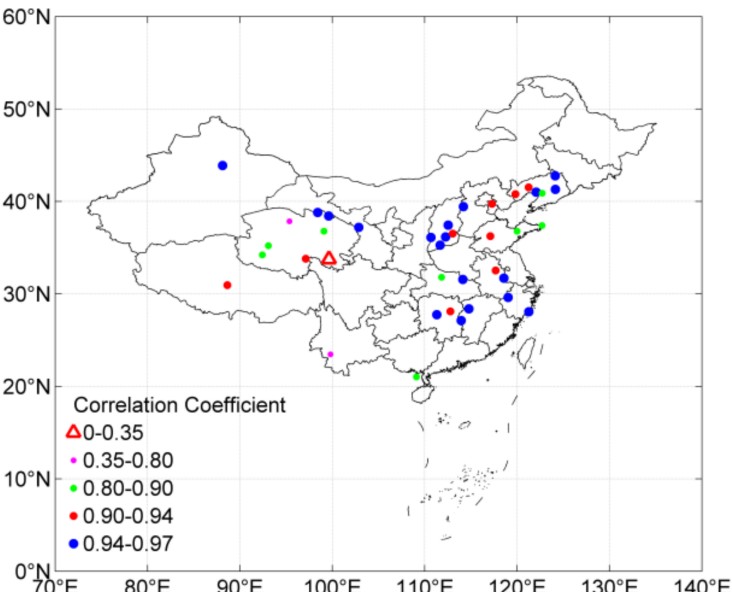

**Figure 2.** Distribution of the correlation coefficients between potential evapotranspiration and evaporation from an evaporation pan during the 1961−2013 period.

Meteorological drought indices are metrics that quantitatively characterize the onset time, intensity change, and end time of drought, serving as an important tool in drought monitoring, prevention, and research. To date, more than 100 meteorological drought indices have been proposed [37,38]. As the SPEI accounts for changes in $E_{TO}$ and has the advantage of multi-scale and multi-spatial comparison [6,7], it is employed widely and was also employed in this study. SPEI can be calculated using the following three steps: (1) calculate $E_{TO}$ using the PM model; (2) calculate the water balance $D_i = P_{RE} - E_{TO}$, that is, the difference between precipitation and $E_{TO}$, and calculate the time series of cumulative water gain/loss, $D_n^k = \sum_{i=0}^{k-1}\left(P_{RE_{n-i}} - E_{TO_{n-i}}\right)$, $n \geq k$, where k represents the time scale (i.e., k months) and n is a given month; (3) obtain SPEI in a manner adapted from the method used for deriving SPI, namely fitting water balance data into a log-logistic distribution to transform the original values to standardized units. During this process, a three-parameter function is used to describe the log-logistic cumulative distribution because $D_i$ may take negative values. Refer to reference [6] for the detailed F(x). A SPEI value of less than −0.5 indicates meteorological drought, and the more negative the value the more severe the drought (the opposite of climate aridity). Drought levels are classified as shown in Table 2.

**Table 2.** Drought severity classification based on SPEI.

| Severity | Type | SPEI Value |
|---|---|---|
| 0 | No drought | $-0.5 <$ SPEI |
| 1 | Mild drought | $-1.0 <$ SPEI $\leq -0.5$ |
| 2 | Moderate drought | $-1.5 <$ SPEI $\leq -1.0$ |
| 3 | Severe drought | $-2.0 <$ SPEI $\leq -1.5$ |
| 4 | Extreme drought | SPEI $\leq -2.0$ |

## 3. Results

### 3.1. Differences between Climate Aridity Changes and Meteorological Drought Changes

Changes in meteorological drought conditions are more dependent on the time scale of analysis than changes in the background climate aridity; thus, the relationship between climate aridity changes and meteorological drought changes varies across different time scales. First, a larger $I_{AI}$ value means a drier climate (Table 1), while a larger value for SPEI means a wetter climate (Table 2). As shown by the multi-year trends of annual-mean and summer-mean $I_{AI}$ and SPEI across China (Figure 3), the two indices generally have opposite trends although, some regions have positive trends of both quantities, demonstrating an increase in climate aridity concurrent with a decrease in drought frequency. The areas where annual-mean $I_{AI}$ and SPEI both showed positive trends are concentrated in the eastern part of the SMTZ (Figure 3a,c); areas of positive trends in summer-mean $I_{AI}$ and SPEI are concentrated in the central and eastern SMTZ (Figure 3b,d). At the same time, areas of positive trends in summer-mean also exsit in monsoon and non-monsoon zone, but the areas are very small (Figure 3b,d).

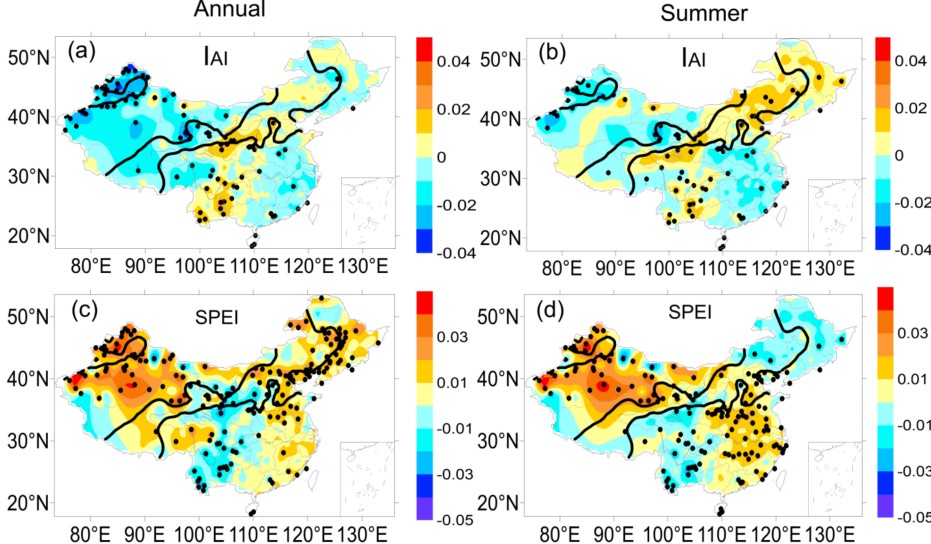

**Figure 3.** Spatial distribution of the multi−year trends of annual (**a**,**c**) and summer (**b**,**d**) mean $I_{AI}$ (**a**,**b**) and SPEI (**c**,**d**) in China from 1961 to 2013. Areas enclosed by the solid contours are semi-arid areas, in which the portion east of 100°E is selected as representative of the summer monsoon transition zone. Black dots in each panel represent stations with a significant correlation between $I_{AI}$ and SPEI for the summer period or the whole year.

Second, as shown by the time series of station-averaged, annual-mean, and summer-mean $I_{AI}$ and SPEI in China (Figure 4), the out-of-phase relationship between the two indices is not evident during some periods; for example, the 1965–1970 period for the annual mean (Figure 4a) and the 1978–1982 and 1987–1994 periods for the summer mean show weak out-of-phase relationships, with respective correlation coefficients of −0.75 (Ra = 0.81), −0.83 (Ra = 0.88), and −0.35 (Ra = 0.70) for these three periods.

In order to know whether there are similar weak out-of-phase relationships between two indices, this study further examines the differences between the two indices on times scales from 1 to 48 months. Figure 5 presents the evolution of the standardized $I_{AI}$ index and SPEI index in sub-zone A of the SMTZ at time scales of 1, 3, 6, 12, 24, and 48 months. It is evident that at one-month, three-month and six-month time scales the two indices undergo different changes, with the cycle of fluctuation being more regular for $I_{AI}$ than for SPEI, and with a weak out-of-phase relationship between the two indices. On time scales longer than 12 months, the two indices fluctuate in a much more similar manner, and a clear out-of-phase relationship is observed. In addition, on short time scales, the years with $I_{AI}$ outliers do not correspond one-to-one to the years with SPEI outliers, such as the outlier

years of 2000 on the one-month time scale in sub-zone A (Figure 5). The phenomena of $I_{AI}$ and SPEI evolution on different time scales in sub-zones B and C are similar to those observed in sub-zone A (not shown).

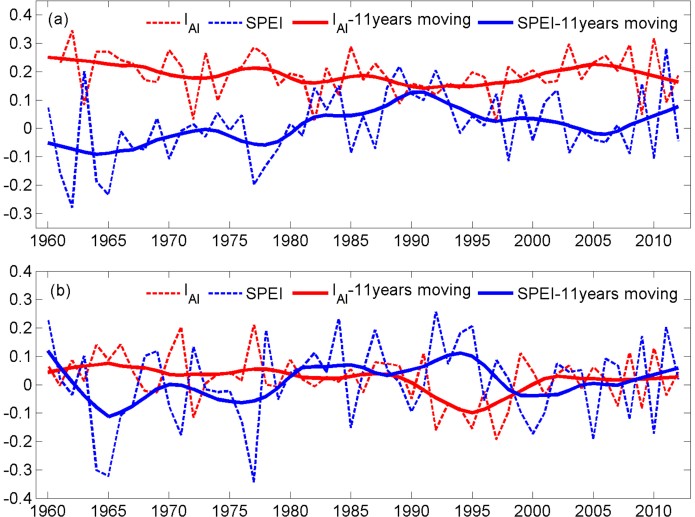

**Figure 4.** Station-averaged annual-mean (**a**) and summer-mean (**b**) $I_{AI}$ and SPEI from 1960 to 2013 (dashed lines) in China. Eleven-year moving averages of $I_{AI}$ and SPEI are shown by the solid lines in each panel (1966 is the first average center).

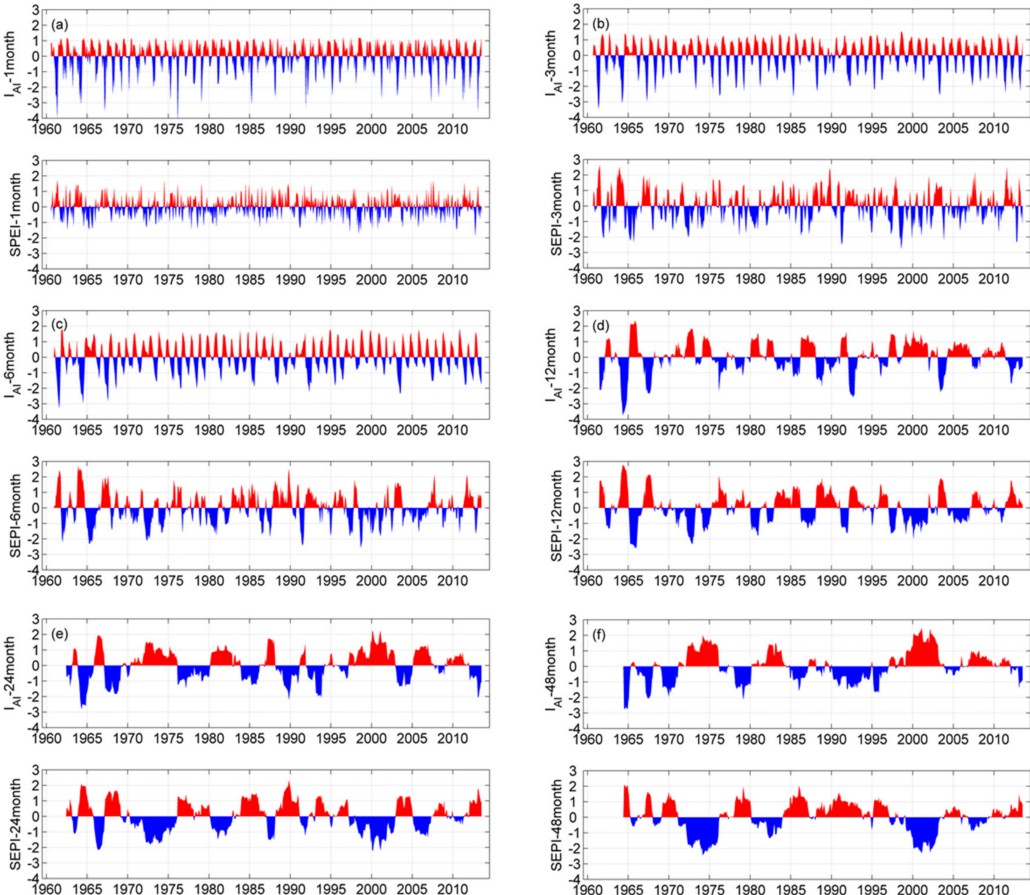

**Figure 5.** Time series of station-averaged $I_{AI}$ and SPEI in sub−zone A of the summer monsoon transition zone on time scales of 1, 3, 6, 12, 24, and 48 months (**a**–**f**), and the upper part of each sub-graph (**a**–**f**) is $I_{AI}$, and the lower part is SPEI.

Figure 6 presents the plots of correlation coefficients between $I_{AI}$ and SPEI versus time scale in the SMTZ (A–C), summer monsoon zone (D), and non-summer-monsoon zone (E) on time scales from 1 to 48 months. Correlation coefficients in the SMTZ (regions A–C) are between 0.40 and 0.60 on the time scales of one, three and six months and between 0.90–1 on the time scales of 12, 24, and 48 months. Sub-zones A and B of the SMTZ have the most similar correlation coefficients of any two regions, and the correlation coefficients in those two sub-zones are lower than in sub-zone C. These results may be because the same weather systems affect the climates of sub-zones A and B, while different systems affect sub-zone C. For example, sub-zones A and B are not only affected by the East Asian summer monsoon and high-latitude westerly jet, but also by the high terrain of the western Tibetan Plateau [20,21,39–41]. In contrast, sub-zone C is mainly affected by the Northeast Cold Vortex and the marine system [42].

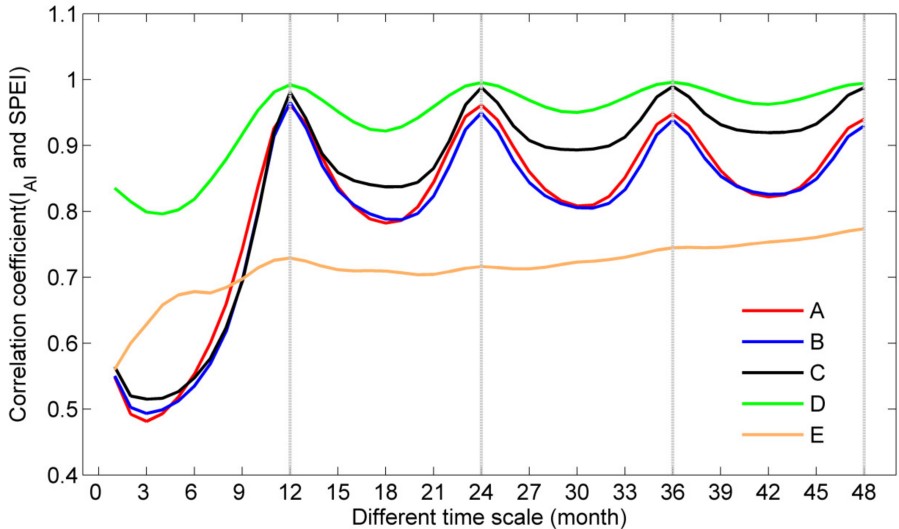

**Figure 6.** Plots of correlation coefficients between $I_{AI}$ and SPEI versus time scales in the different regions shown in Figure 1.

Given that the correlation coefficients are negative, their absolute values are presented here for ease of interpretation, with larger correlation coefficients indicating more similar variation of the two indices. Differences between $I_{AI}$ and SPEI vary from zone to zone. The correlation coefficients between $I_{AI}$ and SPEI in the SMTZ (A–C) and summer monsoon zone (D) show large fluctuations with changes in the time scale, while correlations are more stable in the non-summer-monsoon zone (E). On short time scales (one to six months), the correlation coefficients between $I_{AI}$ and SPEI are smallest (0.40–0.60) in the SMTZ, followed by the non-summer-monsoon zone (0.50–0.70) and the summer monsoon zone (0.75–0.90). This indicates that on short time scales the differences between climate aridity and meteorological drought are the greatest in the SMTZ, while also highlighting the complexity of this zone in terms of meteorological drought and climate aridity. In the SMTZ, between 6 and 12 month time scales, the correlation coefficients between $I_{AI}$ and SPEI increase in a rapid growth, reaching a maximum value (0.95–1) at the 12-month time scale before subsequently declining to a minimum value (0.70–0.80) at the 18-month time scale and rising again to another maximum value (0.95–1) at the 24-month time scale, followed by repeated cycles of similar fluctuation. The pattern of $I_{AI}$–SPEI correlation coefficients with time scale in the summer monsoon zone (D) is similar to that in the SMTZ, although the fluctuations are less dramatic.

The non-summer-monsoon zone (E) shows a distinct evolution in correlation coefficient with time scale compared to the other regions. In this zone, the correlation coefficients on time scales of one to nine months (0.55–0.70) are greater than those of the SMTZ and lower than those of the summer monsoon zone. On time scales beyond nine months, correlation coefficients in the non-summer-monsoon zone (0.70–0.80) are smaller than

those in other zones and show a gradually increasing trend with longer time scales. The main reason for the distinct behavior of the non-summer-monsoon zone is that it is mostly composed of arid and extremely arid areas with a dry background climate and $I_{AI}$ values typically between 0.90 and 1. Therefore, in this region, frequent or continuous meteorological drought events would not cause a dramatic increase in the dryness of the background climate and, in turn, $I_{AI}$ and SPEI would not display a strong out-of-phase relationship.

Figure 7 presents the spatial distribution of the correlation coefficient between $I_{AI}$ and SPEI on time scales of 1, 3, 6, 12, 24, and 48 months. The correlation coefficients increase with increasing time scale in each zone, and are largest in the summer monsoon zone at each time scale. The correlation coefficients on one to six month time scales are smaller in all zones than on 9–12 month time scales. The correlation coefficients on three-month time scales are the smallest in the SMTZ. As shown above, in the SMTZ the inconsistency between changes in background climate dryness and changes in meteorological drought events are most significant on one- to six-month time scales, with a peak in inconsistency at three-month time scales. This was the same result as shown in Figure 6.

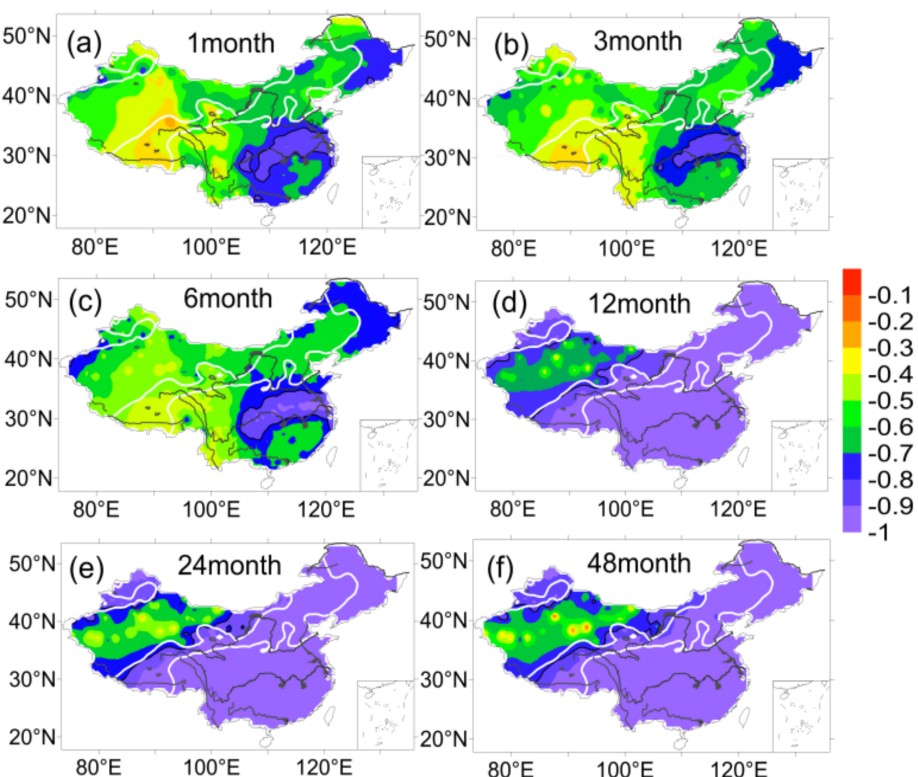

**Figure 7.** Spatial distribution of correlation coefficients between $I_{AI}$ and SPEI on time scales of 1, 3, 6, 12, 24, and 48 months.

### 3.2. Causes of the Differences between Climate Aridity and Meteorological Drought

The causes of the substantial differences between $I_{AI}$ and SPEI on short time scales were examined by analyzing the changes in $I_{AI}$ and SPEI in the historical periods with major meteorological drought events. Based on historical drought records and related research [43–47], historical drought periods were selected as summer 2000, summer 1978, and autumn 1978 for sub-zone A (representative of the SMTZ), the summer monsoon zone (D), and the non-summer-monsoon zone (E), respectively. As typical drought events are intra-seasonal events, changes in $I_{AI}$ and SPEI on time scales of one to six months during typical drought periods were selected for further analysis (Figures 8–10).

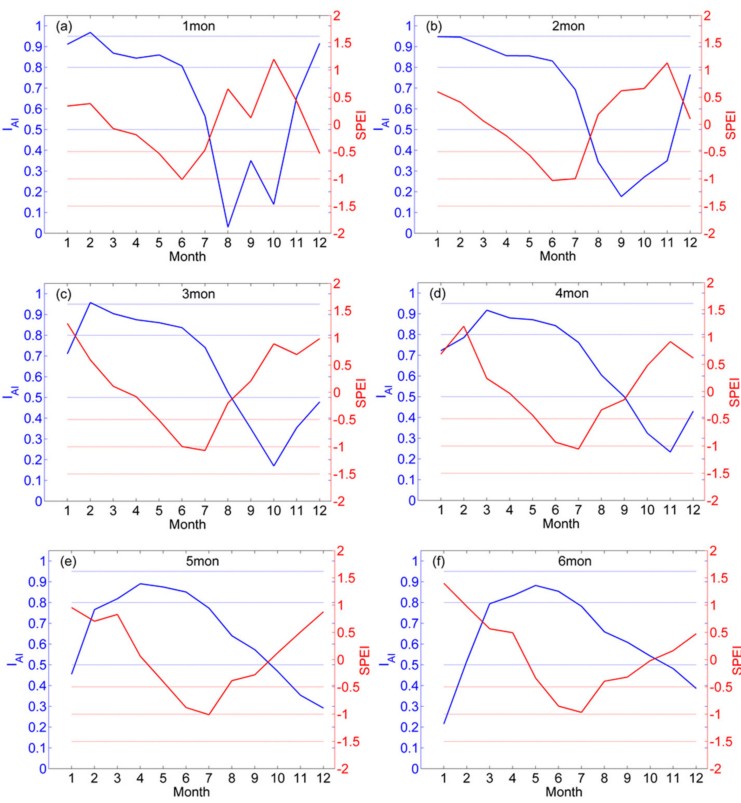

**Figure 8.** Time series of I$_{AI}$ and SPEI on time scales of 1–6 months (**a**–**f**) during 2000 in sub-zone A of the summer monsoon transition zone.

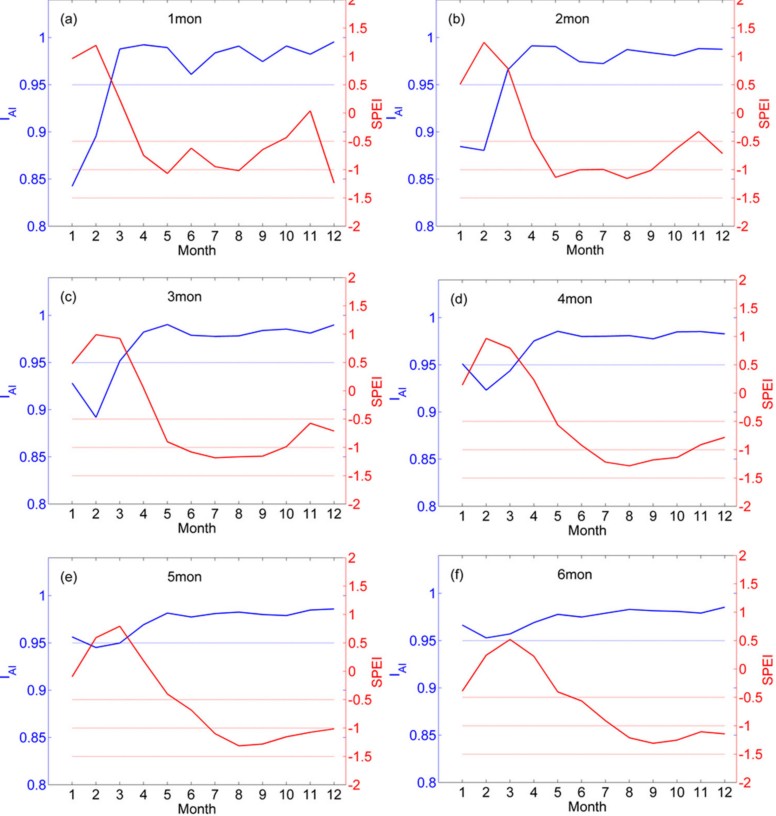

**Figure 9.** Time series of I$_{AI}$ and SPEI on time scales of 1–6 months (**a**–**f**) during 1978 in the non-summer-monsoon zone.

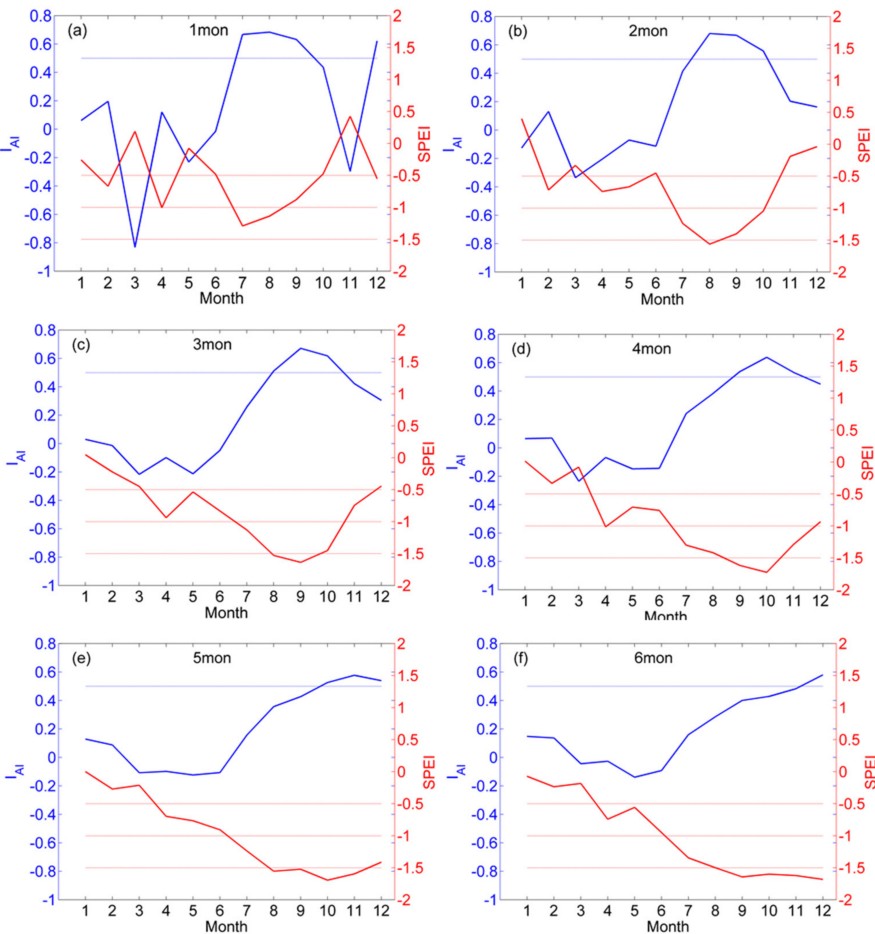

**Figure 10.** Time series of $I_{AI}$ and SPEI on time scales of 1–6 months (**a**–**f**) during 1978 in the summer monsoon zone.

SPEI values < −0.50 indicate an ongoing meteorological drought event (Table 2). Figure 8 presents the changes in $I_{AI}$ and SPEI on time scales of one to six months during 2000 in sub-zone A, which reveals that the two indices vary but are similar on different time scales. On time scales of one month, the occurrence time of SPEI < −0.50 is from early March and then meteorological drought events occur in March, April, May, and July, and the drought is the most significant in May and July. During the periods of meteorological drought, $I_{AI}$ on the one-month time scale reaches its maximum in April and then fluctuates downward, showing rises during June–July, August–September, and October–December. On time scales of two to six months, the occurrence time of SPEI < −0.50 is from early March and then drought events occur in the following months. On time scales of two to six months, $I_{AI}$ reaches its maximum in April and then drops in a relatively smooth manner until September (time scale of two months), October (time scales of three and five months), or November (time scales of four and six months), when it begins to rise again. Thus, on time scales from one to six months, the background climate gradually becomes drier from January until April, when the highest aridity is reached ($I_{AI}$ > 0.95), after which the aridity begins to decrease, with a slight decrease from April to May and a more rapid, fluctuating decrease after May. On time scales of one to six months, during January–April, a meteorological drought deepens in conjunction with the drying of the background climate; however, the drought is most severe in May, rather than April, indicating that drought severity peaks after the maximum dryness of the background climate. After declining between May and June, the SPEI drops to a second minimum in July, which coincides with the rise of $I_{AI}$ between June and July ($I_{AI}$ = 0.71) at a one-month time scale. It is worth noting that when $I_{AI}$ again reaches 0.71 at the end of October no drought occurs

(SPEI = 0.65). A comparison between $I_{AI}$ evolution before July versus before the end of October shows that $I_{AI}$ values are consistently above 0.50 before July and even above 0.85 from February to May, but are consistently below 0.50 from the end of July to the start of October. Thus, in October, the wetter background climate prevents a short period of climate dryness from manifesting in meteorological drought.

The changes in SPEI and $I_{AI}$ at longer time scales show similar characteristics to those described above on one-month time scales. Based on the characteristics of background climate changes during the development of meteorological drought events in May and July, it is found that meteorological drought events are typically preceded by a period of time during which the background climate is dry. That is, the background climate must be relatively dry for a certain period of time before the occurrence of meteorological drought. A similar phenomenon is observed in a typical drought event in 1968 (not shown). The dependence of drought events on the dryness of the background climate accounts for the differences between background climate changes and meteorological drought changes.

Figure 9 shows changes in $I_{AI}$ and SPEI on one- to six-month time scales during 1978 in the non-summer-monsoon zone. The overall results show a similar relationship between the occurrence of meteorological drought and changes in the background climate to that of the SMTZ in 2000; that is, the drying of the background climate precedes the occurrence of meteorological drought. This phenomenon is more prominent in the 1978 non-summer-monsoon zone. On a one-month time scale, meteorological drought mainly occurs from April to September. However, the periods of meteorological drought differ on time scales between two and six months. On time scales of one to two months, the most severe drought occurs in May and the background climate is driest in April; on a three-month time scale the most severe drought occurs in July, and the background climate is driest in May; on time scales of four to six months, the most severe drought occurs in August or September with May remaining the driest month of the background climate. Although the background climate shows a weak trend of increasing moisture, drought still continues to develop in some months, such as April–May on the one- and two-month time scales and May–July on the three-month time scale. This is mainly because after the background climate becomes relatively dry ($I_{AI} > 0.96$), weak fluctuations are insufficient to interrupt the continuity of the dry background climate. The continuation of such a state will eventually lead to the recurrence of meteorological drought, and thus the persistence of drought from April to October.

Figure 10 presents the changes in $I_{AI}$ and SPEI in the summer monsoon zone on time scales of one to six months during 1978, with an out-of-phase relationship between $I_{AI}$ and SPEI. In contrast to the other two zones, the summer monsoon zone exhibits an in-phase relationship between the evolution of meteorological drought and climate aridity. On time scales of one, two, three and four months, meteorological drought is most severe in July, August, September and October, respectively, with the climate aridity reaching its maximum in identical months as meteorological drought. Thus, there is no such out-of-phase relationship between $I_{AI}$ and SPEI as observed in the other two zones. This may be attributed to the fact that the climate aridity in this zone undergoes extreme fluctuations ($-0.83 < I_{AI} < 0.68$ on a one-month time scale) but does not reach very dry values near 1, indicating a relatively wet background climate. When climate aridity in the summer monsoon zone changes, meteorological drought changes accordingly; however, when the background climate of the other two zones reaches a prolonged dry state, a wetting trend may be insufficient to stop the development of meteorological drought.

As shown above, the differences between $I_{AI}$ and SPEI were analyzed from the perspective of the relationship between the occurrence of typical drought events and changes in the background climate. Here, the causes of the above differences are further addressed considering that the two indices have different physical meanings. Climate aridity represents an average state of the background climate. Meteorological drought events, in contrast, are climatic events that occur more stochastically compared to changes in background climate state; changes in meteorological droughts and changes in climate aridity show substantial

differences within a 12-month time scale. The probability density distribution of $I_{AI}$ and SPEI during a period of time reveals the probabilities of background climate states and meteorological drought events during the period.

## 4. Discussion

The above quantitative analysis shows the difference between $I_{AI}$ and SPEI in describing typical drought events and climate background dry and wet changes. It can be seen that the adjustment and change of the climate system is the main reason for the difference between the $I_{AI}$ and SPEI of the summer monsoon transition region, summer monsoon region, and non-summer monsoon. To further reveal its long-term evolution characteristics, we further discuss it from the inter-decadal perspective.

Figures 11–13 present the probability density distribution of standardized $I_{AI}$ and SPEI in sub-zone A (representative of the SMTZ), the non-monsoon zone (E), and the monsoon zone (D), respectively. Overall, across the three zones, the relationship between SPEI and $I_{AI}$ varies with time scale. On time scales in which the differences between SPEI and $I_{AI}$ are the largest (i.e., one, three and six months), the probability density distributions of SPEI and $I_{AI}$ in each zone exhibit inter-decadal variability. On the one-month time scale in sub-zone A (Figure 11g), for example, the SPEI values corresponding to the peak SPEI probability varies greatly across different decades. Peak SPEI probabilities occur for negative SPEI values in the 1960s and 1970s, suggesting higher probabilities of drought occurrence. In contrast, the peak-probability SPEI values are positive in the 1980s, 1990s, and 2010s, and the SPEI probability distribution is shifted rightward compared to earlier decades. In Figure 11a, the peak one-month probabilities of $I_{AI}$ in sub-zone A are similar across decades, and the corresponding $I_{AI}$ values were all positive, with no clear decadal variability observed in the probability density distribution. A similar pattern is also observed on times scales of three and six months, emphasizing that $I_{AI}$ experiences less inter-decadal variability in probability density compared to SPEI. This overall pattern is also observed in the monsoon zone (Figure 12) on time scales of one, three and six months, and in the non-monsoon zone on time scales of one, three, six and twelve months.

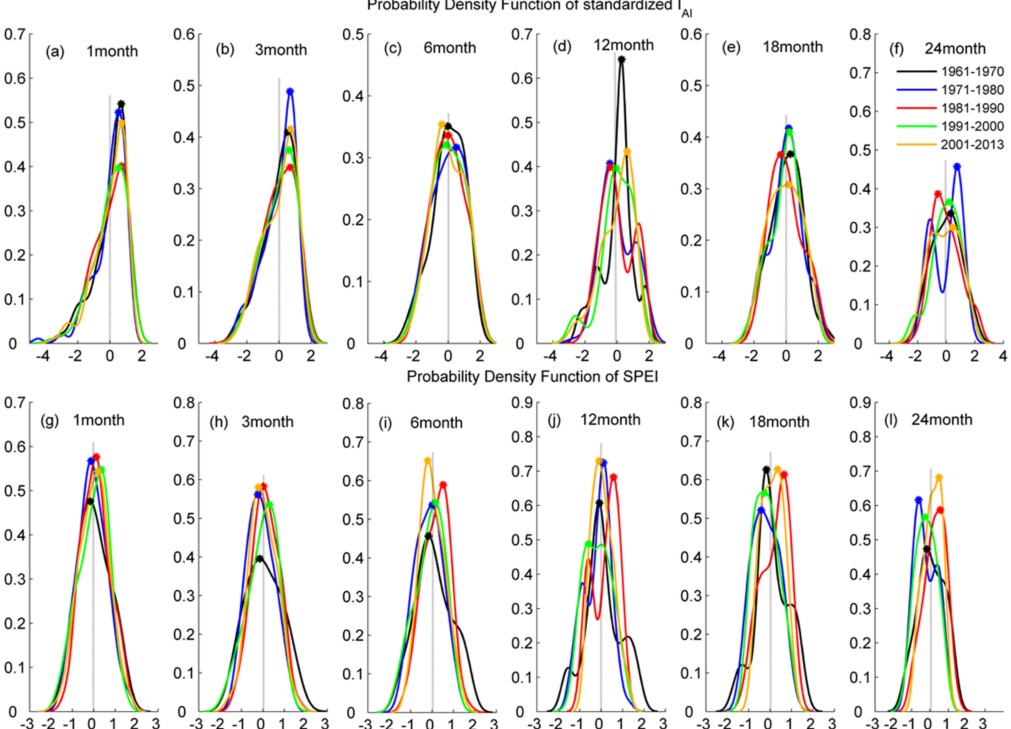

**Figure 11.** Probability density distributions of standardized $I_{AI}$ (**a–f**) and SPEI (**g–l**) in the summer monsoon transition zone (sub-zone A) during each decade.

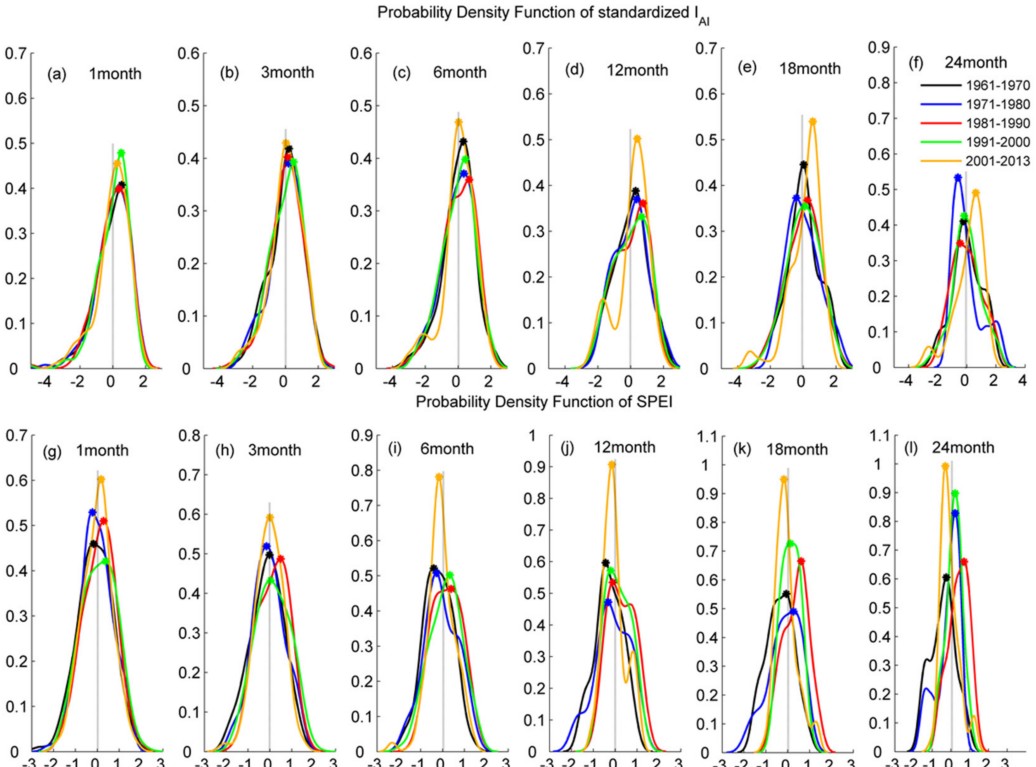

**Figure 12.** Probability density distributions of standardized I$_{AI}$ (**a–f**) and SPEI (**g–l**) in the monsoon zone during each decade.

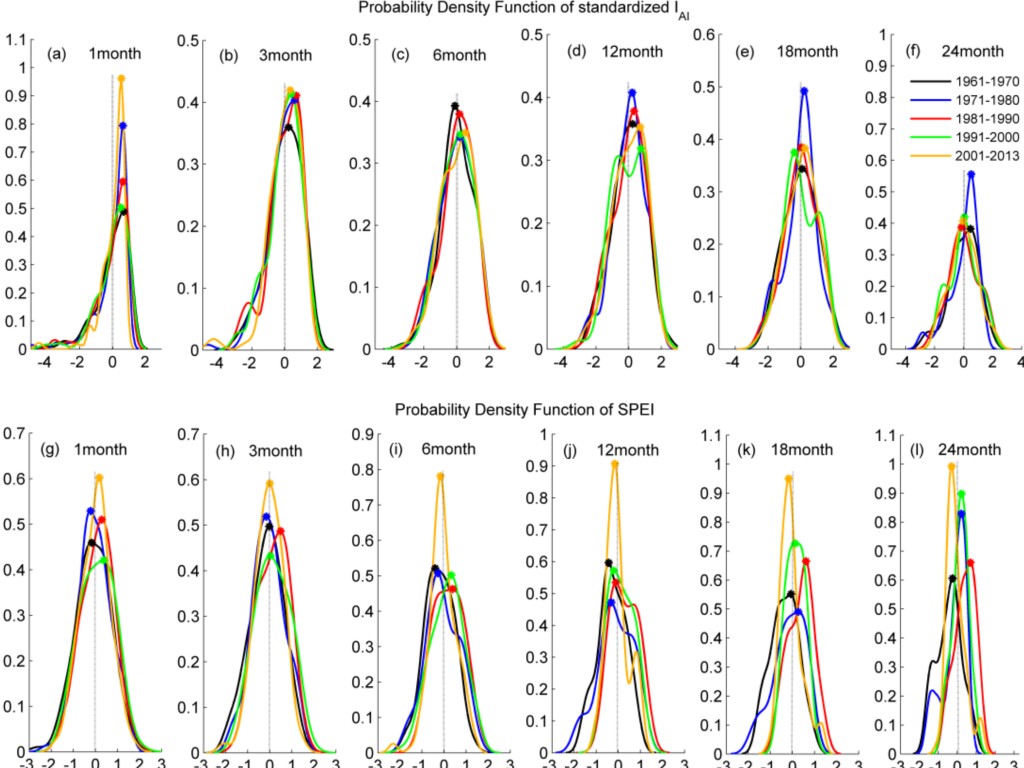

**Figure 13.** Probability density distributions of standardized I$_{AI}$ (**a–f**) and SPEI (**g–l**) in the non-monsoon zone during each decade.

On time scales of 12, 18 and 24 months the three zones exhibit a similar pattern, showing large decadal variability in the probability density distributions of SPEI and $I_{AI}$. Moreover, on these time scales an out-of-phase relationship exists between the two indices. As shown in Figure 11f, for example, the peak-probability SPEI values in the 1960s and 1970s were negative (more drought-like) while the peak-probability $I_{AI}$ values during the same periods (Figure 11i) were positive (more arid climate), and their magnitudes were basically the same as those of the SPEI values. During the 1980s, the peak-probability SPEI values and $I_{AI}$ values were both close to 0. During the 1990s and 2000s, the peak-probability SPEI values were positive (less drought-like) and the peak-probability $I_{AI}$ values in the same periods were negative (less arid climate). These similar inter-decadal characteristics between $I_{AI}$ and SPEI are observed on time scales of 12, 24, and 48 months in the SMTZ and the monsoon zone, but only on time scales of 24 and 48 months in the non-monsoon zone. over long time scales, SPEI contains the average signal of long-term (cumulative) meteorological drought, so changes in drought on these time scales correspond well with changes in climate aridity.

## 5. Conclusions

Previous studies have used climate aridity indices as drought indices to monitor drought on seasonal or shorter time scales. To explore whether climate aridity indices and meteorological drought indices are equally effective for drought monitoring, this study employs a climate aridity index ($I_{AI}$) and meteorological drought index (SPEI) to explore the relationship between the changes in climate aridity and the changes in meteorological drought in China's SMTZ and neighboring regions. Differences between climate aridity and meteorological drought primarily occur on time scales of less than 12 months, and the differences at short time scales are more significant in the SMTZ than in the monsoon zone and the non-monsoon zone. Therefore, using climate aridity to indicate drought events may result in errors when studying short-term droughts, especially in the SMTZ.

Typical drought events were selected to analyze the reasons for the differences between $I_{AI}$ and SPEI at short time scales, and it was found that in both the monsoon transition zone and non-monsoon zone the onset of climate aridity is ahead of the onset of meteorological drought events, with the maximum $I_{AI}$ (representing the driest climate background) appearing earlier than the minimum SPEI (representing the most severe meteorological drought event). In the SMTZ, if the climate changes from a semi-arid state to an arid state and remains arid ($0.80 < I_{AI} < 1$), meteorological drought events will occur and undergo rapid and continuous aggravation. The onset of the most severe climate aridity is two to seven months (mostly three to four months) ahead of the onset of the most severe drought events, until the climate returns to a semi-arid state, when the meteorological drought starts to be alleviated rapidly. The above phenomenon also occurs in the non-monsoon zone, but with two distinct patterns: (1) meteorological drought events are rapidly and continuously aggravated only when the $I_{AI}$ is greater than about 0.975, and (2) the onset of the most severe climate aridity is ahead of the onset of the most severe drought events by only three to four months, which is a shorter period compared to the monsoon transition zone. The above phenomenon, however, is not obvious in the monsoon zone. These findings indicate that $I_{AI}$ can serve as a predictor of the onset of meteorological drought events, especially in the SMTZ, but it fails to characterize the progression of meteorological drought events well. Therefore, this result is of great significance for drought prediction and early warning.

The inconsistency between $I_{AI}$ and SPEI occurs due to two reasons. First, the occurrence of a meteorological drought is not synchronous with the drying of the background climate. After the background climate becomes relatively dry (such as $0.96 < I_{AI} < 1$), weak fluctuations are insufficient to interrupt the continuity of the generally dry climate state. The continued dry state of climate eventually results in increasingly severe meteorological droughts, or the recurrence of equally severe droughts, and the persistence of drought on long time scales. Second, climate aridity represents an average state of the background climate over a long time period, whereas meteorological drought events are stochastic

climate events and are more unpredictable, so the two indices may differ to some extent on short time scales.

**Author Contributions:** Conceptualization, H.Z., L.Z. and Q.Z.; formal analysis, H.Z. and L.Z.; funding acquisition, Q.Z. and L.Z.; investigation, H.Z., L.Z., Q.L., X.Y. and L.W.; project administration, H.Z.; visualization, H.Z.; writing—original draft, H.Z.; writing—review and editing, H.Z., L.Z., Q.Z., Q.L., X.Y. and L.W. All authors have read and agreed to the published version of the manuscript.

**Funding:** This study was funded by the National Natural Science Foundation of China (Grant Nos. 42230611 and 41875020), the Natural Science Foundation of Gansu Province (Grant Nos. 20JR10RA446, 20JR10RA804) and the Ten-talent Plan of Gansu Meteorological Bureau.

**Data Availability Statement:** Not applicable.

**Conflicts of Interest:** The authors declare no conflict of interest.

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
