# Peer review of "Analysis of the Difference between Climate Aridity Index and Meteorological Drought Index in the Summer Monsoon Transition Zone"

_remotesensing, doi:10.3390/rs15051175_

Round 1
Reviewer 1 Report
It is better to make the abstract shorter.
Author Response
Response to Reviewer 1 Comments
Dear reviewer,
Thank you very much for your review. According your comments, some revison have been done as follows:
Point 1: It is better to make the abstract shorter.
Response 1: I am sorry that the abstract part is slightly longer, and the meaning of some sentence is repeated. Some repetitive sentences have been deleted and sentences have been simplified, such as ' Differences between the two indices... Explore ', ' when the meteorological drought. SMTZ.', 'when the data are analyzed', 'therefore, the two indices differ to some extent on short time scales', and so on. The number of words in the abstract has been reduced from 381 to 310, and the content is as follows:
The summer monsoon transition zone (SMTZ) in China represents an unusual land type with an agro-pasture ecotone, and it is a climate-sensitive region. Changes in climate aridity and changes in meteorological drought are mutually related yet fundamentally different. In this study potential evapotranspiration(ETO) is calculated using Penman-Monteith, based on China’s national meteorological stations data from 1961–2013. An ETO-based climate aridity index (IAI) and ETO-based standardized precipitation evapotranspiration index (SPEI) are used as the metrics for climate aridity and meteorological drought, respectively. The result shows a significant difference between climate aridity and meteorological drought in the SMTZ, compared with monsoon and non-monsoon Zone. This difference varies on different time scales (1-48months), and the greatest differences between IAI and SPEI are on seasonal and monthly scales (1-12 months), but lower at longer time scales (> 12months). The first reason for the difference is that desynchroneity of meteorological drought and the background climate. After the background climate becomes relatively arid state (such as 0.96 < IAI < 1) from semi-arid state (0.5 < IAI < 0.8), the continued arid state with weak IAI fluctuations eventually results in increasingly severe meteorological droughts, or the recurrence of equally severe droughts with drastic reduction. So that the onset of the most severe climate aridity is 2–7 months (mostly 3–4 months) ahead of the onset of the most severe drought events, until the climate returns to a semi-arid state. Second, climate aridity represents the average state of the background climate over a long time period and changes gently, while meteorological droughts are stochastic climate events and change drasticly. These findings indicate that IAI can serve as a predictor of the onset of meteorological drought events, especially in the SMTZ, but it fails to well characterize the progression of meteorological drought events. Therefore, this result is of great significance for drought prediction and early warning.

Reviewer 2 Report
In this manuscript, the authors work to show differences in the meteorological drought index (SPEI) and climate aridity index (IAI) on short time scales in different regions of China, with a focus on the summer monsoon transition zone (SMTZ) in northern China. After selecting methods to compute climate aridity and meteorological drought, they use station data from 1961 to 2013, obtained from four regions in China, to show that on 1-6-month time scales, there are substantial differences in the climate aridity index and meteorological drought index. These differences are especially pronounced in the summer monsoon transition zone, a region of transition between the monsoon zone to the south, and the non-monsoon zone to the northwest. Overall, it is a good job to explain the differences between climate aridity and meteorological drought, and using station data to show how the two quantities differ. But some Comments are as follows:
1. The abstract part is slightly longer, and the meaning of the sentence is repeated. Please consider whether ' Differences between the two indices... Explore 'in line 18-19, ' when the meteorological drought... SMTZ. ' in line 32-33.
2. SPEI is a widely cited drought index, and the calculation program has a general program. If the algorithm used in this paper is its universal algorithm, it is recommended to delete the algorithm introduction of line 204-216, and retain the introduction of the steps and significance of SPEI, and the division of drought levels in table 2.
3. The authors competently describe the data used in the study and their quality control of the data. However, it is somewhat unclear whether the data from every station used are available from 1961-2013. I think that is the case, based on the results, but the authors should briefly clarify this.
4. A short justification should be provided for why the part of the SMTZ at 100 East is selected as representative of the SMTZ.
5. The number of decimal places in the text should be consistent, such as two digits after the decimal point
Author Response
Response to Reviewer 2 Comments
Dear reviewer,
Thank you very much for your review. According your comments, some revison have been done as follows:
Point 1: The abstract part is slightly longer, and the meaning of the sentence is repeated. Please consider whether some sentences can be delated, such as ' Differences between the two indices... Explore 'in line 18-19, ' when the meteorological drought... SMTZ. ' in line 32-33 .
Response 1: Those sentence 'Differences between the two indices... Explore ' in line 18-19 and 'when the meteorological drought... SMTZ. ' in line 32-33 of the first draft have been delated.
Point 2: SPEI is a widely cited drought index, and the calculation program has a general program. If the algorithm used in this paper is its universal algorithm, it is recommended to delete the algorithm introduction of line 204-216, and retain the introduction of the steps and significance of SPEI, and the division of drought levels in table 2.
Response 2: The detalied algorithm introduction of line 204-216 of the first draft has been delated, and a literature for the algorithm has been added in lines 201-202:
Lines 204-205 : Refer to literature [6] for the detailed F(x).
Point 3: The authors competently describe the data used in the study and their quality control of the data. However, it is somewhat unclear whether the data from every station used are available from 1961-2013. I think that is the case, based on the results, but the authors should briefly clarify this.
Response 3: The information 'the data from every station used are available from 1961-2013’ has been added in lines 116-118:
Lines 116-118: Using these selection criteria, 661 stations were selected, and the data from every station used are available, with a monitoring period from 1961 to 2013.
Point 4: A short justificatiaon should be provided for why the part of the SMTZ at 100 East is selected as representative of the SMTZ.
Response 4: A short justification has been provided for why the part of the SMTZ at 100 East is selected as representative of the SMTZ in lines 121-125:
Lines 121-125: Because the west of the semi-arid area at 100 ° E is located in the Tibet Plateau, which is mainly affected by the South Asian high, while the east of the semi-arid area at 100 ° E is almost located at the northern edge of the East Asian summer monsoon (mainly the southwest and southeast summer monsoon), the east of the semi-arid area(0.5 < IAI < 0.8) at 100 ° E is selected as the representative area of the SMTZ.
Point 5: The number of decimal places in the text should be consistent, such as two digits after the decimal point
Response 5: All the number of decimal places have been revised in this artile.

Reviewer 3 Report
Review of Remote Sensing 2211005, “Analysis of the Difference between Climate Aridity Index and Meteorological Drought Index in the Summer Monsoon Transition Zone.”
The article describes a quantitative comparison of an index of background climate aridity and an index of meteorological drought. At annual and longer time scales, such indices often behave comparably. Some specific differences between two such indices at sub-annual time scales in the SMTZ in China are identified. For example, in the SMTZ, meteorological droughts usually follow (rather than cause) a period of background aridity (whereas background climate wetness tends to preclude the onset of meteorological drought). At time scales longer than 1 year, the two indices tend to exhibit an out-of-phase relationship, although this varies across space. Interestingly, the same out-of-phase relationship exists between the indices when examining interdecadal variability at annual and longer time scales. The spatial and temporal differences between the two indices suggest that indexes of climate aridity may be poorly suited to analyses of drought at seasonal or shorter time scales.
The paper is well-written and thorough, and I don’t question the results. I have no major comments or suggestions. I include minor comments and suggestions below.
Minor comments:
1) As I said above, I don’t question the results, but I think the paper would benefit from a clearer statement (either in the Introduction or Conclusions or both) of the specific value of this study. The authors hint at what I think is the main point, that Climate Aridity Indices are probably not well-suited to analyze short-term drought events. But is that a common problem? Are investigators frequently using long-term aridity measures to analyze short-term droughts? If so, do your results help shed any light on any possible misinterpretations from the past? Or do these results indicate something about drought predictability? This isn’t a huge problem, but I found myself wondering “what can we do with this information?”
2) With the possible exception of Figure 1, I found all the figures difficult to read based on their size. This was especially true for the multi-panel figures. Any way to increase the legibility of the figures would help the reader. I realize this may be a matter for the publisher.
3) Figure 1: region “C” appears to be labeled instead with an “O”
4) line 256: the number “12” is repeated unnecessarily

Author Response
Response to Reviewer 3 Comments
Point 1: As I said above, I don t question the results, but I think the paper would benefit from a clearer statement (either in the Introduction or Conclusions or both) of the specific value of this study. The authors hint at what I think is the main point, that Climate Aridity Indices are probably not well-suited to analyze short-term drought events. But is that a common problem? Are investigators frequently using long-term aridity measures to analyze short-term droughts? If so, do your results help shed any light on any possible misinterpretations from the past? Or do these results indicate something about drought predictability? This isn t a huge problem, but I found myself wondering what can we do with this information?
Response 1: Thank you for your advance. I'm sorry that I didn't make it clear in the introduction and conclusion. In fact, many studies have used the indicators of climate dryness and wetness to study drought, which has been mentioned in the introduction. The specific contents are as follows (in lines 87-98 of the article):
Lines 84-94:‘Similarly, some studies still use the index (surface humidity index or relative humidity index) which essentially represents climate dryness as the drought index to study drought on the seasonal and shorter monthly scales. Wang et al. [28] used the relative moisture index to investigate the temporal and spatial distribution of seasonal drought events in southwestern China; Zhou et al. [29] used aridity to define drought level and investigated meteorological and climate characteristics during drought periods; Ma et al.[27] also used the relative moisture index to measure changes in northeastern China drought trends during May–September from 1961 to 2009. Yao et al. [30] used the relative moisture index as a spring drought indicator to study drought in southwestern China; Huang et al. [18] used the moisture index to calculate the frequency of extreme drought in northwestern China.’
Our results do help shed some light on possible misinterpretations from the past. We find that when the climate background becomes the driest, the most serious drought events do not occur at the same time, but lag behind the climate background for 2-7 months in the SMTZ, and 3-4 months in the non-monsson zone. Therefore, the dry-wet change of climate background can be used as an early warning signal of drought.
Some corresponding explanations have been reviesed and added in the introduction and conclusion. See lines 84-86, 94-95, 472-474, 490-494 for details:
Lines 84-86: Similarly, some studies still use the index (surface humidity index or relative humidity index) which essentially represents climate dryness as the drought index to study drought on the seasonal and shorter monthly scales.
Lines 94-95: But is it reliable to study drought by using the index that essentially represents climate dryness and wetness?
Lines 469-471: Therefore, using climate aridity to indicate drought events may result in errors when studying short-term droughts, especially in the SMTZ.
Lines 487-491: These findings indicate that IAI can serve as a predictor of the onset of meteorological drought events, especially in the SMTZ, but it fails to well characterize the progression of meteorological drought events. Therefore, this result is of great significance for drought prediction and early warning.
Point 2: With the possible exception of Figure 1, I found all the figures difficult to read based on their size. This was especially true for the multi-panel figures. Any way to increase the legibility of the figures would help the reader. I realize this may be a matter for the publisher.
Response 2: All the pictures in the article have been enlarged. If the publisher has requirements in the later typesetting, I will try my best to cooperate.
Point 3: Figure 1: region C appears to be labeled instead with an O
Response 3: Figure 1 has been replayed in the article, thank you very much for your advance.
Figure 1. Distribution of representative monitoring stations in the summer monsoon transition zone (A–C), monsoon zone (D), and non-monsoon zone (E).
Point 4: line 256: the number 12 is repeated unnecessarily
Response 4: The number 12 has been delated in line 245:
Line 242:Figure 5 presents the evolution of the standardized IAI index and SPEI index in sub-zone A of the SMTZ at time scales of 1, 3, 6, 12, 24, and 48 months.
